# DIFFUSION LARGE LANGUAGE MODELS FOR BLACK-BOX OPTIMIZATION

## ABSTRACT

Offline black-box optimization (BBO) aims to find optimal designs based solely on an offline dataset of designs and their labels. Such scenarios frequently arise in domains like DNA sequence design and robotics, where only a few labeled data points are available. Traditional methods typically rely on task-specific proxy or generative models, overlooking the in-context learning capabilities of pre-trained large language models (LLMs). Recent efforts have adapted autoregressive LLMs to BBO by framing task descriptions and offline datasets as natural language prompts, enabling direct design generation. However, these designs often contain bidirectional dependencies, which left-to-right models struggle to capture. In this paper, we explore diffusion LLMs for BBO, leveraging their bidirectional modeling and iterative refinement capabilities. This motivates our *in-context denoising* module: we condition the diffusion LLM on the task description and the offline dataset, both formatted in natural language, and prompt it to denoise masked designs into improved candidates. To guide the generation toward high-performing designs, we introduce *masked diffusion tree search*, which casts the denoising process as a step-wise Monte Carlo Tree Search that dynamically balances exploration and exploitation. Each node represents a partially masked design, each denoising step is an action, and candidates are evaluated via expected improvement under a Gaussian Process trained on the offline dataset. Our method, *dLLM*, achieves state-of-the-art results in few-shot settings on design-bench, with code available here.

## 1 INTRODUCTION

Designing new objects or entities to optimize specific properties is a fundamental challenge across domains such as DNA sequence design and robotics Trabucco et al. (2022). Since querying these properties is often expensive Hamidieh (2018); Angermüller et al. (2020); Barrera et al. (2016); Sample et al. (2019), recent efforts have focused on offline settings that avoid online evaluations. This gives rise to offline black-box optimization (BBO) (Kim et al., 2025), where the goal is to discover high-performing designs using only an offline dataset of candidates and their associated labels. The challenge is particularly pronounced when only a few labeled data points are available.

Traditional methods typically either (1) train a proxy model (Trabucco et al., 2021; Yuan et al., 2023; Hoang et al., 2024; Yu et al., 2021; Fu & Levine, 2021), which provides explicit gradient guidance over existing designs, or (2) train a conditional generative model (Kumar & Levine, 2020; Krishnamoorthy et al., 2023; Brookes et al., 2019) that takes a target value as input to directly generate high-scoring designs. However, such task-specific and data-limited methods are prone to out-of-distribution generalization issues and overlook the general in-context learning capabilities of pre-trained large language models (LLMs) (Brown et al., 2020), which offer a versatile alternative.

Recent efforts have adapted autoregressive LLMs for BBO by framing task descriptions and offline datasets as natural language prompts. These models generate improved candidate designs directly through in-context learning (Yang et al., 2024; Zhang et al., 2023; Liu et al., 2024; Nie et al., 2024; Veličković et al., 2024; Novikov et al., 2025). However, many design tasks involve bidirectional dependencies—for instance, a DNA unit is influenced not only by its prefix but also by its suffix. Since autoregressive LLMs generate sequences in a left-to-right fashion, they struggle to fully model such dependencies, especially in high-dimensional and complex design spaces.

In this paper, we explore diffusion LLMs for BBO. These models offer two key advantages: bidirectional modeling and iterative refinement, making them well-suited for design tasks with complex dependencies. As shown in Figure 1, we introduce an **in-context denoising** module: the task description and the offline dataset are formatted as natural language prompts. These prompts, followed by an instruction to generate improved designs, are then provided to the diffusion LLM, which iteratively denoise masked designs into improved candidates.

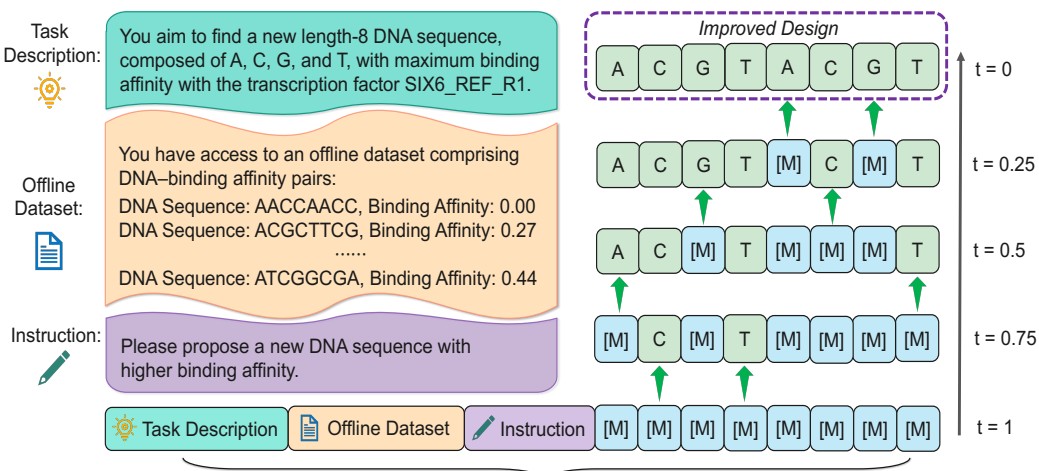

Figure 1: In-context denoising.

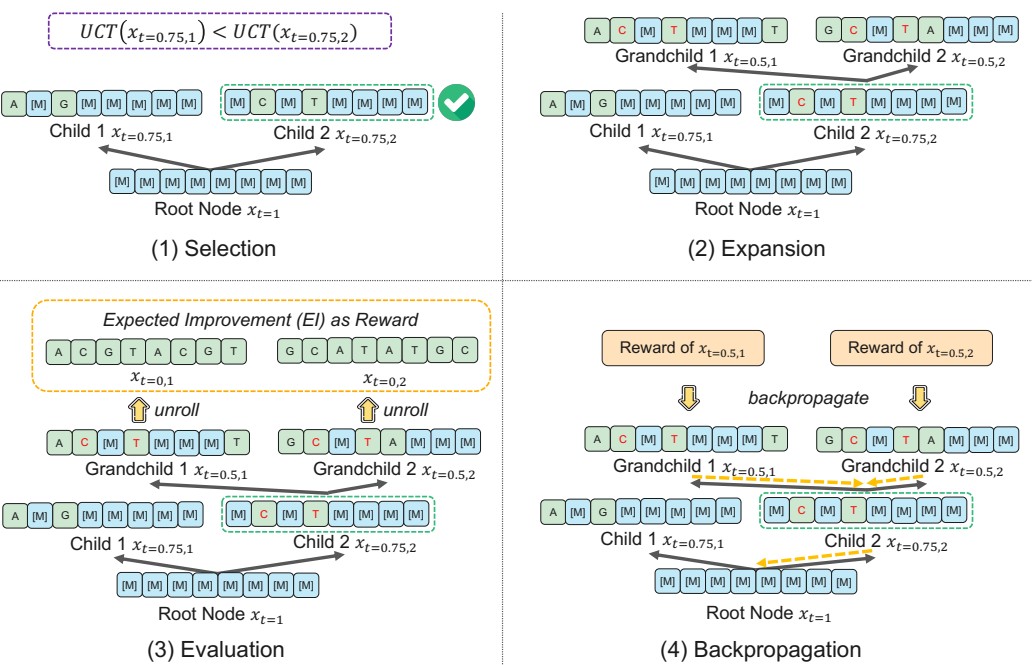

Figure 2: Masked diffusion tree search.

*In-context denoising* alone is often insufficient for guiding generation toward high-performing designs. To address this, we introduce a **masked diffusion tree search** module (Figure 2), which casts the denoising process as a step-wise Monte Carlo Tree Search (MCTS): each masked design is a tree node, and each denoising step is an action. The search proceeds through four stages to dynamically balance exploration and exploitation: (1) *Selection*: starting from the root node $x_{t=1}$

(fully masked design), traverse the tree using the UCT score (Kocsis & Szepesvári, 2006), which balances the exploitation of high-value nodes with the exploration of less-visited ones, to identify a promising node (e.g., select $\boldsymbol{x}_{t=0.75,2}$ over $\boldsymbol{x}_{t=0.75,1}$ due to its higher UCT score); (2) *Expansion*: apply diffusion LLM denoising to the selected node (e.g., $\boldsymbol{x}_{t=0.75,2}$) to generate new child nodes (e.g., $\boldsymbol{x}_{t=0.5,1}$, $\boldsymbol{x}_{t=0.5,2}$); (3) *Evaluation*: unroll each child node (e.g., $\boldsymbol{x}_{t=0.5,1}$) into complete designs (e.g., $\boldsymbol{x}_{t=0,1}$) and estimate its reward using expected improvement under a Gaussian Process; (4) *Backpropagation*: propagate the reward signal up the tree (e.g., $\boldsymbol{x}_{t=0.5,1} \rightarrow \boldsymbol{x}_{t=0.75,2} \rightarrow \boldsymbol{x}_{t=1}$), thereby updating ancestors and informing future decisions.

To summarize, our contributions are three-fold:

- To the best of our knowledge, we are the first to explore diffusion LLMs for BBO, leveraging their bidirectional modeling and iterative refinement capabilities.

- We propose *in-context denoising*, which conditions diffusion LLMs on natural language task descriptions and offline datasets to denoise masked designs into improved candidates.

- We introduce *masked diffusion tree search*, which casts diffusion LLM denoising as a step-wise Monte Carlo Tree Search that adaptively balances exploration and exploitation.

## 2 PRELIMINARIES

### 2.1 OFFLINE BLACK-BOX OPTIMIZATION

Offline black-box optimization (BBO) aims to find the optimal design $\boldsymbol{x}^*$ that maximizes an unknown objective function $f(\cdot)$:

$$\boldsymbol{x}^* = \arg\max_{\boldsymbol{x}} f(\boldsymbol{x}). \tag{1}$$

To tackle this, we assume access to an offline dataset $\mathcal{D} = \{(\boldsymbol{x}_i, y_i)\}_{i=1}^{N}$ containing $N$ data points, where each $\boldsymbol{x}_i \in \mathbb{R}^d$ represents a $d$-dimensional design and $y_i$ is its associated score. In many practical scenarios, only a few labeled data points are available; thus, we focus on the few-shot regime where $N$ is small (e.g., $N = 10$).

A common method is to fit a deep neural network (DNN) model to approximate $f(\cdot)$ in a supervised manner Chen et al. (2023), and then leverage it to guide the design optimization. However, in the few-shot setting, Gaussian Process (GP) models are often more effective than learned DNNs due to the limited data and their ability to provide principled uncertainty estimates.

Specifically, we adopt a GP model (MacKay et al., 1998) with a radial basis function (RBF) kernel $K(\cdot, \cdot)$. Given an offline dataset $\boldsymbol{X} \in \mathbb{R}^{N \times d}$ and its corresponding labels $\boldsymbol{y} \in \mathbb{R}^N$, we model a new design $\boldsymbol{x}$ by computing the predictive posterior distribution:

$$\hat{y}(\boldsymbol{x}) \sim \mathcal{N}(\mu(\boldsymbol{x}), \sigma^2(\boldsymbol{x})). \tag{2}$$

### 2.2 DIFFUSION LARGE LANGUAGE MODELS

Large language models (LLMs) are widely used in natural language processing tasks. In contrast to conventional autoregressive models (ARMs) such as GPT Brown et al. (2020), diffusion LLMs Nie et al. (2025b); Gong et al. (2025); Arriola et al. (2025) have emerged as a promising alternative, achieving competitive results with models of comparable size. Compared to ARMs, diffusion LLMs offer two key advantages: (1) the ability to leverage bidirectional context, and (2) an iterative sampling process that naturally scales at test time.

A diffusion LLM defines a distribution $p_{\boldsymbol{\theta}}(\boldsymbol{x}_0)$ via a two-step process: a *forward corruption process* and a *reverse denoising process*. In the forward process, tokens in the original sequence $\boldsymbol{x}_0$ are progressively masked until, at $t = 1$, the entire sequence is fully masked. For any intermediate timestep $t \in (0, 1)$, the sequence $\boldsymbol{x}_t$ is partially masked, with each token masked independently with probability $t$ and left unmasked with probability $1 - t$. The reverse process then reconstructs the original data distribution by iteratively recovering masked tokens as $t$ decreases from 1 to 0.

At the core of a diffusion LLM is a *mask predictor*, a parametric model $p_{\boldsymbol{\theta}}(\cdot|\boldsymbol{x}_t)$ that takes the partially masked sequence $\boldsymbol{x}_t$ as input and predicts all masked tokens (denoted by M) simultaneously.

The model is trained by minimizing a cross-entropy loss computed only over the masked positions:

$$\mathcal{L}(\theta) = -\mathbb{E}_{t, \boldsymbol{x}_0, \boldsymbol{x}_t} \left[ \frac{1}{t} \sum_{i=1}^{L} \mathbf{1}[\boldsymbol{x}_t^i = \mathrm{M}] \log p_\theta(\boldsymbol{x}_0^i | \boldsymbol{x}_t) \right], \tag{3}$$

where $\boldsymbol{x}_0$ is sampled from the training corpus, $t$ is drawn uniformly from the interval $[0, 1]$, and $\boldsymbol{x}_t$ is generated by the forward masking process. The indicator function $\mathbf{1}[\cdot]$ ensures that only masked tokens contribute to the loss. Once trained, the mask predictor enables the simulation of the reverse denoising process, producing the model distribution $p_\theta(\boldsymbol{x}_0)$ as the final marginal distribution.

## 3 METHOD

Algorithm 1 outlines our overall method, combining *in-context denoising* (Sec. 3.1) with *masked diffusion tree search* (Sec. 3.2).

### 3.1 IN-CONTEXT DENOISING

**In-Context Prompt**  We construct the prompt to diffusion LLM by concatenating the following:

- **Task Description**: specifies the meaning and format of the design, the associated label, and the optimization objective (maximize or minimize). For example:

> Task Description
>
> You aim to find a new length-8 DNA sequence, composed of A, C, G, and T, with maximum binding affinity with the transcription factor SIX6 REF R1.

- **Offline Dataset**: provides a few-shot set of design–label pairs. For example:

> Offline Dataset
>
> You have access to an offline dataset comprising DNA–binding affinity pairs:
> DNA Sequence: GGCCGGCC, Binding Affinity: 0.00
> DNA Sequence: GCCCTTCG, Binding Affinity: 0.27
> ......
> DNA Sequence: GTGGGCGA, Binding Affinity: 0.44

- **Instruction**: prompts the model to generate improved candidates. For example:

> Instruction
>
> Please propose a new DNA sequence with higher binding affinity.

**Denoising**  The resulting structured prompt (*Task Description, Offline Dataset, Instruction*) is provided to the diffusion LLM, which then iteratively denoises masked candidates into higher-performing designs. Specifically, let $\boldsymbol{x}_t$ denote a masked design at step $t$, starting from $t = 1$ for the fully masked design. Given a sampling interval $\Delta t$, the next step is $s = t - \Delta t$. At each step, the diffusion LLM predicts all masked tokens in $\boldsymbol{x}_t$ to produce a complete candidate $\hat{\boldsymbol{x}}_0$. We then remask the least confident tokens in $\hat{\boldsymbol{x}}_0$, using a masking ratio of $s/t$, to form the new child node $\boldsymbol{x}_s$ (Nie et al., 2025b). We restrict sampling to valid logits only. For example, in DNA sequence design the diffusion LLM is limited to the alphabet {A, C, G, T}, while in numerical design tasks only digits and basic symbols ("0–9, -, .") are permitted. Any additional invalid designs (e.g., strings that do not form a valid number) are excluded. The final $\boldsymbol{x}_0$ is returned as the desired design.

### 3.2 MASKED DIFFUSION TREE SEARCH

We introduce *masked diffusion tree search* to enable more effective guided generation. This module frames the denoising process as step-wise decision-making: each node in the tree corresponds to

---

**Algorithm 1 Diffusion Large Language Models for Black-Box Optimization**

---

**Input:** Offline dataset $\mathcal{D} = \{(\boldsymbol{x}_i, y_i)\}_{i=1}^N$, pre-trained diffusion LLM $p_{\boldsymbol{\theta}}(\cdot)$, # of iterations $M$

1: Fit an RBF-kernel GP $f(\cdot)$ on $\mathcal{D}$.
2: Construct the in-context prompt (*Task Description, Offline Dataset, Instruction*) as in Sec. (3.1).
3: Initialize the search tree with the root node $\boldsymbol{x}_1$.
4: **for** iter $= 1$ to $M$ **do**
5:     **Selection:** Traverse the tree from $\boldsymbol{x}_1$ using the UCT score in Eq. (4) and select node $\boldsymbol{x}_t$.
6:     **Expansion:** Expand $\boldsymbol{x}_t$ to generate diverse children $\boldsymbol{x}_{s,i}$ by completion and remasking.
7:     **Evaluation:** Unroll each child $\boldsymbol{x}_{s,i}$ into full designs and compute reward $r_{s,i}$ via Eq. (5).
8:     **Backpropagation:** Propagate $r_{s,i}$ to all ancestors using Eqs. (6) and (7).
9: **end for**
10: Return fully denoised designs $\boldsymbol{x}_0$ with the highest EI scores among explored candidates.

---

a partially masked design $\boldsymbol{x}_t$, which can be further denoised to generate a new child node $\boldsymbol{x}_{s,i}$, where $s < t$ and $i$ indexes the $i$-th child. We follow the standard MCTS steps: *selection*, *expansion*, *evaluation*, and *backpropagation*, to dynamically balance exploration and exploitation.

**Selection**      At each iteration, we traverse the tree starting from the parent node $\boldsymbol{x}_u$ and select a child node $\boldsymbol{x}_{s,i}$ using the UCT score (Kocsis & Szepesvári, 2006):

$$\text{UCT}(\boldsymbol{x}_{s,i}) = V(\boldsymbol{x}_{s,i}) + \omega \, p_{\boldsymbol{\theta}}(\boldsymbol{x}_{s,i}|\boldsymbol{x}_u)\sqrt{\frac{\log N(\boldsymbol{x}_u)}{N(\boldsymbol{x}_{s,i})}}. \tag{4}$$

Here, $V(\boldsymbol{x}_{s,i})$ is the current value estimate of node $\boldsymbol{x}_{s,i}$, encouraging exploitation of promising branches. The model likelihood $p_{\boldsymbol{\theta}}(\boldsymbol{x}_{s,i}|\boldsymbol{x}_u)$ guides the search toward more probable unmasking steps (Silver et al., 2016). $N(\boldsymbol{x}_u)$ denotes the visit count of the parent node, while $N(\boldsymbol{x}_{s,i})$ is the visit count of the child node. The exploration term $\sqrt{\frac{\log N(\boldsymbol{x}_u)}{N(\boldsymbol{x}_{s,i})}}$ therefore encourages selecting less-visited nodes. The coefficient $\omega = 1.0$ controls the trade-off between exploitation and exploration. This process is repeated until reaching a leaf node that is not yet expanded.

**Expansion**      Once a node $\boldsymbol{x}_t$ is selected for expansion, the diffusion LLM is used to generate design completions. New child nodes $\boldsymbol{x}_{s,i}$ are then constructed by remasking tokens according to the ratio $s/t$. By sampling different completions according to the branching factor, we can generate diverse children $\boldsymbol{x}_{s,i}$ from the same parent.

**Evaluation**      To evaluate the new child node $\boldsymbol{x}_{s,i}$, we perform multiple denoising runs to obtain $J$ fully unmasked designs $\boldsymbol{x}_{0,i}^{(j)}$. We then compute their rewards using the Expected Improvement (EI) criterion from the Gaussian Process:

$$\text{EI}(\boldsymbol{x}_{0,i}^{(j)}) = \left(\mu(\boldsymbol{x}_{0,i}^{(j)}) - f_{\text{best}}\right)\Phi(z_j) + \sigma(\boldsymbol{x}_{0,i}^{(j)})\phi(z_j), \quad \text{where} \quad z_j = \frac{\mu(\boldsymbol{x}_{0,i}^{(j)}) - f_{\text{best}}}{\sigma(\boldsymbol{x}_{0,i}^{(j)})}. \tag{5}$$

where $\mu(\cdot)$ and $\sigma(\cdot)$ are the GP predictive mean and standard deviation, $\Phi$ and $\phi$ are the standard normal CDF and PDF, and $f_{\text{best}}$ is the best score in the offline dataset. Finally, we set the node's reward to the average EI: $r_{s,i} = \frac{1}{J}\sum_{j=1}^J \text{EI}(\boldsymbol{x}_{0,i}^{(j)})$.

**Backpropagation**      After obtaining the reward $r_{s,i}$ for $\boldsymbol{x}_{s,i}$, we backpropagate it through the entire trajectory. For each ancestor node $\boldsymbol{x}_\tau$ with $\tau > s$, we update its visit count and value estimate:

$$N_{\text{new}}(\boldsymbol{x}_\tau) = N_{\text{old}}(\boldsymbol{x}_\tau) + 1, \tag{6}$$

$$V_{\text{new}}(\boldsymbol{x}_\tau) = \frac{V_{\text{old}}(\boldsymbol{x}_\tau) \cdot N_{\text{old}}(\boldsymbol{x}_\tau) + r_{s,i}}{N_{\text{new}}(\boldsymbol{x}_\tau)}. \tag{7}$$

Finally, we return fully denoised designs $\boldsymbol{x}_0$ that achieve high EI scores from explored candidates.

Our *masked diffusion tree search module* is proposed as a practical UCT-based heuristic for exploring high-dimensional design spaces. While classical UCT offers guarantees under idealized conditions, extending such analysis to our setting—where expansions come from a diffusion LLM and rewards rely on a GP-based EI—would require strong additional assumptions and is beyond the scope of this work. Instead, we focus on its empirical behavior, with extensive ablations in Sec. (4.5) and sensitivity studies in Sec. (4.6) showing that our module is effective and robust across tasks.

## 4 EXPERIMENTS

We conduct extensive experiments to evaluate the effectiveness of our *dLLM* in few-shot settings of design-bench. Specifically, we benchmark against established baselines in Sec. (4.4), assess the contribution of each component in Sec. (4.5), and analyze hyperparameter sensitivity in Sec. (4.6).

### 4.1 BENCHMARKS

**Datasets** Following (Nguyen et al., 2023), we evaluate on two continuous and two discrete tasks. The continuous tasks are: **(1) Ant Morphology (Ant)** Brockman et al. (2016), which involves optimizing a 60-D ant morphology for fast crawling; and **(2) D'Kitty Morphology (D'Kitty)** Ahn et al. (2020), which requires optimizing a 56-D D'Kitty morphology for the same objective. The discrete tasks are: **(1) TF Bind 8 (TF8)** Barrera et al. (2016), which requires designing an 8-length DNA sequence to maximize binding activity with the SIX6_REF_R1 transcription factor; and **(2) TF Bind 10 (TF10)** Barrera et al. (2016), which extends this to a 10-length sequence. We construct few-shot settings by uniformly sampling **10** examples from the offline datasets.

**Evaluation** For each generated design, we adopt the oracle described in the *Design-Bench Benchmark Tasks* Trabucco et al. (2022) for evaluation. In line with prior work Trabucco et al. (2021), we evaluate 128 candidates per method and report the maximum (i.e., $100^{\text{th}}$ percentile) normalized ground-truth score. The normalized score $y_n$ is computed as $y_n = \frac{y - y_{\min}}{y_{\max} - y_{\min}}$, where $y$ denotes the design's raw score, and $y_{\min}$ and $y_{\max}$ are the minimum and maximum scores across the full unobserved dataset. To provide a more comprehensive comparison, we also report the mean and median ranks of competing methods, as well as the median normalized scores.

### 4.2 BASELINES

We benchmark our method against a broad range of established baselines.

**Proxy-based methods** These methods focus on learning a reliable proxy to guide design updates. **(1) Grad (mean)**: uses the GP predictive mean $\mu(\boldsymbol{x})$ as a proxy predictor, and updates offline designs using its gradient. **(2) Grad (EI)**: uses the expected improvement $\text{EI}(\boldsymbol{x})$ in Eq. (5) as a proxy predictor, and updates offline designs using its gradient. **(3) COMs** (Trabucco et al., 2021): applies conservative objectives by lower-bounding a neural proxy's predictions on out-of-distribution designs, followed by proxy gradient ascent. **(4) ICT** (Yuan et al., 2023): employs a rotating pseudo-labeler and co-teaching strategy among three proxies, followed by meta-learned sample reweighting. **(5) MATCH-OPT** (Hoang et al., 2024): bounds performance gaps via proxy–gradient mismatch, improving proxy quality and optimization performance. **(6) UniSO-T** (Tan et al., 2025): trains a sequence-to-sequence autoregressive proxy to predict labels which are encoded as tokens, and uses the trained proxy to guide design generation.

**Generative model-based methods** These methods directly model the distribution of promising designs using VAE, GAN, autoregressive, or diffusion models. **(1) CbAS** (Brookes et al., 2019): trains a VAE and progressively adapts it to high-scoring designs. **(2) ExPT** (Nguyen et al., 2023): pretrains a transformer-based VAE on diverse synthetic functions and performs in-context generation to sample improved designs. **(3) MIN** (Kumar & Levine, 2020): uses a GAN to model the inverse mapping from score to design, then queries improved designs by conditioning on high scores. **(4) BONET** (Mashkaria et al., 2023): trains an autoregressive model on trajectories from low- to high-scoring samples and unrolls it to generate candidates. **(5) OPRO** (Yang et al., 2024): feeds design–label histories into autoregressive LLMs to directly sample new designs. We adopt LLaMA3-8B-Instruct (Dubey et al., 2024) due to its comparable size to LLaDA-8B-Instruct. **(6) GTG** (Yun

Table 1: Experimental results in 100-th percentile normalized scores on four tasks for comparison.

| Method | Ant Morphology | D'Kitty Morphology | TF Bind 8 | TF Bind 10 | Rank Mean | Rank Median |
|---|---|---|---|---|---|---|
| $\mathcal{D}$(**best**) | 0.565 | 0.884 | 0.439 | 0.467 | – | – |
| Grad-mean | $0.628 \pm 0.019$ | $0.922 \pm 0.006$ | $0.700 \pm 0.103$ | $0.603 \pm 0.033$ | 10.3/16 | 9.5/16 |
| Grad-EI | $0.620 \pm 0.022$ | $\underline{0.939} \pm \underline{0.001}$ | $0.701 \pm 0.109$ | $0.570 \pm 0.036$ | 9.5/16 | 11.0/16 |
| COMs | $0.629 \pm 0.022$ | $0.934 \pm 0.002$ | $0.681 \pm 0.086$ | $0.596 \pm 0.038$ | 10.0/16 | 9.5/16 |
| ICT | $0.639 \pm 0.015$ | $0.925 \pm 0.009$ | $0.752 \pm 0.050$ | $0.605 \pm 0.024$ | 8.0/16 | 8.5/16 |
| MATCH-OPT | $0.619 \pm 0.018$ | $0.920 \pm 0.004$ | $0.696 \pm 0.050$ | $0.598 \pm 0.065$ | 11.5/16 | 10.5/16 |
| UniSO-T | $\underline{0.642} \pm \underline{0.010}$ | $\underline{0.939} \pm \underline{0.022}$ | $0.844 \pm 0.033$ | $0.631 \pm 0.018$ | $\underline{2.8}$/16 | $\underline{2.5}$/16 |
| CbAS | $0.594 \pm 0.036$ | $0.910 \pm 0.007$ | $0.738 \pm 0.026$ | $0.614 \pm 0.008$ | 11.3/16 | 11.5/16 |
| ExPT | $0.641 \pm 0.021$ | $0.935 \pm 0.004$ | $0.826 \pm 0.036$ | $0.640 \pm 0.014$ | 3.3/16 | 3.5/16 |
| MIN | $0.588 \pm 0.065$ | $0.892 \pm 0.009$ | $0.776 \pm 0.011$ | $0.567 \pm 0.010$ | 13.5/16 | 14.5/16 |
| BONET | $0.638 \pm 0.031$ | $0.927 \pm 0.003$ | $0.782 \pm 0.115$ | $0.595 \pm 0.035$ | 8.0/16 | 7.0/16 |
| ORPO | $0.596 \pm 0.014$ | $0.878 \pm 0.009$ | $0.778 \pm 0.033$ | $0.499 \pm 0.033$ | 13.0/16 | 14.0/16 |
| GTG | $0.597 \pm 0.021$ | $0.904 \pm 0.002$ | $0.795 \pm 0.079$ | $0.630 \pm 0.035$ | 8.8/16 | 9.0/16 |
| DDOM | $0.589 \pm 0.016$ | $0.902 \pm 0.002$ | $0.760 \pm 0.064$ | $0.615 \pm 0.033$ | 11.3/16 | 11.5/16 |
| CMA-ES | $0.603 \pm 0.142$ | $0.725 \pm 0.002$ | $0.811 \pm 0.057$ | $0.628 \pm 0.044$ | 9.5/16 | 8.5/16 |
| MCTS-transfer | $0.633 \pm 0.017$ | $0.933 \pm 0.020$ | $\underline{0.848} \pm \underline{0.023}$ | $\underline{0.638} \pm \underline{0.008}$ | 4.3/16 | 4.5/16 |
| **dLLM**$_{(\text{ours})}$ | $\mathbf{0.652 \pm 0.030}$ | $\mathbf{0.942 \pm 0.012}$ | $\mathbf{0.876 \pm 0.038}$ | $\mathbf{0.642 \pm 0.013}$ | **1.0/16** | **1.0/16** |

et al., 2024): learns a conditional diffusion model on synthetic trajectories from offline data to guide designs toward high-scoring regions. **(7) DDOM** (Krishnamoorthy et al., 2023): trains a conditional diffusion model on offline datasets and applies classifier-free guidance to obtain candidates.

We also compare against other methods: **(1) CMA-ES** (Hansen, 2006): adapts a covariance matrix toward high-scoring regions. **(2) MCTS-transfer** (Wang et al., 2024): adaptively generates improved designs by using Monte Carlo tree search to iteratively divide and explore subspaces.

### 4.3 IMPLEMENTATION DETAILS

We adopt the pre-trained dLLM LLaDA-8B-Instruct as our base model (Nie et al., 2025b) and fit a Gaussian Process (GP) model with an RBF kernel on the offline dataset. The search tree is limited to a depth of 4 with a branching factor of 5. The total number of tree search steps, denoted by $M$ in Algorithm 1, is set to 16, while the number of fully unmasked designs $J$ is set to 5. The sampling temperature of the dLLM is set to 1.0. The full in-context prompts are provided in Appendix A.2. All experiments are conducted on a single NVIDIA A100 GPU with 80GB of memory. In terms of runtime, a full search episode takes approximately 30 minutes on Ant and 40 minutes on TF8.

### 4.4 RESULTS AND ANALYSIS

We report the 100-th percentile normalized scores in Table 1, highlighting the best and second-best results in **bold** and underline, respectively, and summarize mean/median ranks across tasks. Results for the 50-th percentile are provided in the Appendix A.3.

Our key observations are: **(1)** *dLLM* achieves the highest average and median rank among 15 baselines. It ranks first on all four tasks, showing consistent gains in both continuous and discrete design spaces. **(2)** Pure gradient-based methods (Grad-mean, Grad-EI) perform markedly worse, indicating that proxies alone cannot reliably identify good designs. A powerful generative model like dLLM coupled with tree search, is crucial for effective exploration of design space. **(3)** Among proxy-based methods, UniSO-T performs best, likely due to leveraging a pre-trained T5 model with strong unsupervised knowledge. However, it is limited to regression use. **(4)** ExPT is the strongest among generative baselines, benefiting from pre-training on synthetic functions. Yet, it lags behind *dLLM*, which draws on broader pre-trained knowledge. This is further supported by DDOM (a diffusion model without external pre-training), which performs worse. **(5)** ORPO, which combines LLaMA3-8B-Instruct with the GP model feedback, delivers weaker results than *dLLM*. Its left-to-right AR modeling by nature restricts its ability to fully capture bidirectional dependencies within designs, particularly in TF8 and TF10.

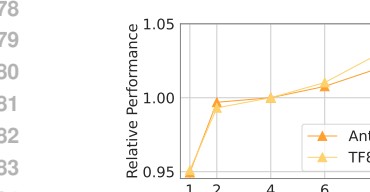 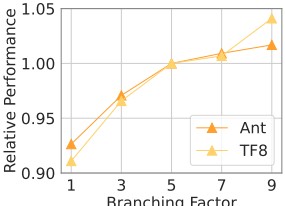 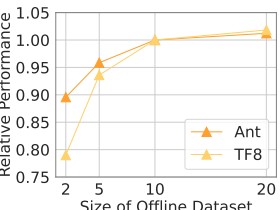

Figure 3: Sensitivity to different tree depths.

Figure 4: Sensitivity to different branching factors.

Figure 5: Sensitivity to different offline dataset sizes.

## 4.5 ABLATION STUDIES

We ablate *dLLM* by replacing or removing its key modules, as detailed in Table 2.

**Vanilla diffusion**    We replace the diffusion LLM with a continuous diffusion model trained directly on the offline dataset (Krishnamoorthy et al., 2023). This causes a clear performance drop, especially on discrete sequence tasks, since the model is trained on few-shot samples without the broad pre-trained knowledge of dLLM, making it difficult to generalize to unseen designs.

**w/o MDTS**    We remove the *masked diffusion tree search* (MDTS) module and let the diffusion LLM to generate candidates directly from the in-context prompt, followed by selecting the best candidates based on the GP predictor. This results in a substantial decrease in performance across all tasks, highlighting the importance of tree-based exploration in navigating the design space.

**w/o template**    We ablate the prompt construction by removing the task description, retaining only the offline dataset and the instruction as input. Without this explicit task framing, the model's performance drops across all tasks, suggesting that clear semantic guidance in the prompt helps the diffusion LLM better interpret the objective and generate more aligned candidates.

Table 2: Ablation studies.

| Method | Ant | D'Kitty | TF8 | TF10 |
|---|---|---|---|---|
| Vanilla diffusion | $0.590 \pm 0.018$ | $0.897 \pm 0.008$ | $0.764 \pm 0.068$ | $0.603 \pm 0.014$ |
| w/o MDTS | $0.604 \pm 0.003$ | $0.892 \pm 0.001$ | $0.798 \pm 0.012$ | $0.503 \pm 0.013$ |
| w/o template | $0.630 \pm 0.005$ | $0.934 \pm 0.015$ | $0.846 \pm 0.027$ | $0.633 \pm 0.017$ |
| **dLLM**$_{(ours)}$ | $\mathbf{0.652 \pm 0.030}$ | $\mathbf{0.942 \pm 0.012}$ | $\mathbf{0.876 \pm 0.038}$ | $\mathbf{0.642 \pm 0.013}$ |

Our ablations demonstrate the necessity of each component. We further test *cls-free guidance* (Nie et al., 2025a), which yields similar results but doubles inference cost, so it is not adopted.

## 4.6 HYPERPARAMETER SENSITIVITY

We conduct sensitivity analysis on two representative tasks: Ant and TF8. All results are reported as relative performance, where the performance is normalized by that of our default configuration.

**Tree Depth**    In *masked diffusion tree search*, the tree depth corresponds to the number of denoising steps to reach a fully unmasked design. The default is $4$, and we vary it over $\{1, 2, 4, 6, 8\}$. As shown in Figure 3, performance improves steadily with increasing depth, reflecting the benefit of iterative refinement in dLLM denoising.

**Branching Factor**    We next examine the effect of the branching factor, i.e., the number of candidate children expanded at each node. The default is $5$, and we vary it across $\{1, 3, 5, 7, 9\}$. As shown in Figure 4, larger branching factors generally yield better performance, highlighting the advantage of broader exploration. It is worth mentioning that dLLM naturally supports both sequential scaling via tree depth and parallel scaling via branching factor, though gains are ultimately limited by the GP model's epistemic uncertainty.

**Size of Offline Dataset**    We then vary the offline dataset size, which influences both the GP predictor training and the offline dataset in the in-context prompt. The default size is $10$, and we test $\{2, 5, 10, 20\}$. As shown in Figure 5, performance drops sharply when only 2 samples are available,

as both the GP and the prompt lack sufficient knowledge. Beyond this extreme case, performance improves steadily and stabilizes with more data. In Appendix A.4, we further study the effect of using different in-context prompts, and find that our method is robust to prompt variations.

**dLLM**  In our main experiments, we adopt LLaDA-8B-Instruct (Nie et al., 2025b) as the diffusion LLM backbone. We additionally test MMaDA-8B-MixCoT (Yang et al., 2025), a model of comparable scale with unified multimodal pretraining. The results are quite close on Ant ($0.649 \pm 0.023$ with MMaDA vs. $0.652 \pm 0.030$ with LLaDA) and TF8 ($0.862 \pm 0.027$ with MMaDA vs. $0.876 \pm 0.038$ with LLaDA), indicating that our method is robust to the choice of diffusion LLM backbone.

## 5  RELATED WORK

**LLMs for BBO**  LLMs have recently gained attention in the field of BBO due to their strong pattern recognition and in-context learning capabilities Song et al. (2024). Two primary research lines have emerged: (1) using LLMs to predict the property $y$ of a given black-box design $x$, typically with fine-tuning; and (2) prompting LLMs directly—without parameter updates—to generate promising designs $x$. Bidirectional models such as T5 encoders (Raffel et al., 2020) have been widely used in the first line due to their ability to capture bidirectional dependencies: Nguyen et al. (2024); Tang et al. (2024) investigate the use of LLM embeddings for regression and demonstrate their effectiveness in high-dimensional design spaces. Tan et al. (2025) further introduces a unified string-based representation to encode both metadata and design values. Autoregressive LLMs have also been explored: Zhao et al. (2024) analyzes their decision boundaries on binary classification tasks, while Nguyen & Grover (2024) augments them with molecular embeddings and property predictors to autoregressively predict molecular properties.

In the second line, autoregressive LLMs have been employed directly as optimizers in BBO. Zhang et al. (2023) frames task descriptions and labeled designs as prompts to guide design generation, and Liu et al. (2024) uses LLMs to select parents for crossover and mutation in an evolutionary fashion. Beyond general design tasks, program synthesis has emerged as a specific and promising domain, as code can be naturally described in language and evaluated via compilers. FunSearch (Veličković et al., 2024) and AlphaEvolve (Novikov et al., 2025) treat code as a design space and use LLM-guided evolutionary strategies to optimize for target objectives, achieving strong empirical results. Our work follows the second line: we leverage the in-context learning ability of LLMs to generate improved designs. However, unlike prior work that predominantly uses autoregressive models (Zhang et al., 2023), we investigate diffusion LLMs, which offer bidirectional modeling and iterative sampling capabilities, making them especially suitable for black-box design problems.

**Monte Carlo Tree Search**  MCTS has been explored in BBO to improve sample efficiency and search quality. LA-MCTS (Wang et al., 2020) guides black-box optimizers by learning space partitions and concentrating sampling within promising local regions. Wang et al. (2024) uses MCTS to adaptively construct and refine the search space for new tasks by transferring knowledge from similar source tasks. Liu et al. (2025) applies a reward-guided MCTS framework within ESM3 to discover protein sequences that fold into a given backbone structure.

MCTS has also been explored in the context of diffusion models, where the denoising process is reinterpreted as a sequential tree-based search—each node representing a partially denoised sample and each action corresponding to one denoising step. Yoon et al. (2025a) introduce Monte Carlo Tree Diffusion, and Yoon et al. (2025b) further accelerate it through parallel rollouts and trajectory coarsening. Zhang et al. (2025) incorporate a hybrid MCTS strategy into the diffusion denoising pipeline, achieving scalable test-time inference These studies primarily focus on planning tasks, such as maze navigation. Jain et al. (2025) propose a tree-based method that samples from a reward-aligned target distribution using a pre-trained diffusion model, demonstrating strong results in text-to-image generation and language modeling tasks. Closest to our work, Tang et al. (2025) propose an MCTS framework built on top of diffusion language models for molecular optimization. However, their model operates over SMILES representations, limiting its generality to small molecules and lacking the in-context learning capabilities of LLMs.

## 6 CONCLUSION

Autoregressive LLMs have recently been applied to black-box design generation but struggle to capture bidirectional dependencies. In this work, we explored diffusion LLMs for offline BBO, leveraging their bidirectional modeling and iterative refinement capabilities. We introduced two key modules: *in-context denoising*, which conditions diffusion LLMs on task descriptions and offline datasets to refine masked designs, and *masked diffusion tree search*, which formulates denoising as a Monte Carlo Tree Search to balance exploration and exploitation. Together, these modules enable our method to achieve state-of-the-art results on design-bench in the few-shot setting. Our results demonstrate the potential of diffusion LLMs as general-purpose optimizers and open new directions for data-efficient design generation in scientific and engineering domains.

## REPRODUCIBILITY STATEMENT

To ensure reproducibility, we provide comprehensive implementation details in Section 4.3. In addition, the source code necessary to replicate our experiments is made available here.

## ETHICS STATEMENT

Our proposed method, *dLLM*, has the potential to accelerate scientific discovery in areas such as biomedicine and robotics by enabling more effective optimization of complex design spaces. These benefits could foster meaningful advancements across multiple domains. At the same time, as with many powerful optimization tools, there is also a risk of misuse. In particular, the ability to generate and refine high-performing designs might be misapplied in harmful contexts, for instance to accelerate the development of biological weapons or other malicious technologies. To mitigate such risks, it is crucial that appropriate safeguards, oversight, and regulatory frameworks be established to ensure that the method is used responsibly and strictly for beneficial purposes.

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

# A   APPENDIX

## A.1   LLM USAGE

We used large language models solely to assist in polishing the writing of this paper. In particular, LLMs were employed to refine wording and check grammar for clarity and readability. No part of the method design, implementation, or experimental results analysis relied on LLM assistance.

## A.2   IN-CONTEXT PROMPT

For clarity and reproducibility, we provide the in-context prompts used in our experiments.

---

**In-Context Prompt for Ant**

You are a helpful optimization assistant that will help us generate a new robot morphology design. The goal is to optimize the morphological structure of a simulated robot: Ant from OpenAI Gym. For Ant Morphology, we need to optimize the morphology of a quadruped robot to run as fast as possible.

For each design, we have the following information, that is the 60 continuous values of the morphology parameters which are grouped into 4 legs, each leg has 15 parameters: x : the x-coordinate of the hip joint, y : the y-coordinate of the hip joint, z : the z-coordinate of the hip joint, a : the angle of the hip joint, b : the angle of the thigh joint, c : the angle of the ankle joint, hip center : the center of the hip joint, hip range : the range of the hip joint, thigh center : the center of the thigh joint, thigh range : the range of the thigh joint, ankle center : the center of the ankle joint, ankle range : the range of the ankle joint, hip size : the size of the hip joint, thigh size : the size of the thigh joint, ankle size : the size of the ankle joint

You are given the following existing designs and their corresponding performance scores:

Robot Morphology Design: [0.01, 0.01, 0.01, 15.03, 15.02, 58.04, -24.89, 4.57, -16.5, 28.86, 61.91, 23.22, 0.2, 0.21, 0.38, 0.01, 0.0, 0.01, 12.08, 27.69, 108.36, -0.24, 8.41, 5.2, 31.64, 41.59, 18.12, 0.22, 0.19, 0.36, 0.0, 0.0, -0.01, -14.78, 2.05, 195.67, 8.69, 2.64, -3.88, 29.51, 32.58, 19.1, 0.19, 0.2, 0.4, 0.01, -0.0, -0.02, 35.92, -12.99, -61.83, -20.47, 4.67, -24.25, 29.19, 22.72, 19.29, 0.22, 0.21, 0.43], Performance Score: -386.9
...
Robot Morphology Design: [-0.01, -0.0, 0.0, 7.64, -7.49, 1.64, 10.86, 5.26, 25.13, 28.7, 33.16, 21.44, 0.19, 0.2, 0.44, -0.02, -0.01, 0.01, 8.55, 7.52, 78.95, -11.6, 5.98, 13.54, 29.48, 41.69, 19.64, 0.2, 0.23, 0.38, -0.0, -0.01, 0.0, -13.43, -2.04, 170.37, 9.33, 6.51, -0.43, 31.12, 54.72, 22.07, 0.19, 0.18, 0.4, 0.0, -0.0, -0.0, 5.91, -20.12, -92.43, 17.22, 6.47, 10.4, 30.79, 45.95, 18.89, 0.18, 0.21, 0.44], Performance Score: 165.33

Please propose a new robot morphology design to maximize the performance score. Each feature should be a float number. Each number should be rounded to two decimal places.

---

864
865

**In-Context Prompt for D'Kitty**

866
867
868
869

You are a helpful optimization assistant that will help us generate a new robot morphology design. The goal is to optimize the morphological structure of a simulated robot: D'Kitty. For D'Kitty Morphology, we aim to optimize the body and leg structure of the robot to maximize its locomotion ability to navigate the robot to a fixed location.

870
871
872
873
874
875
876
877
878

For each design, we have the following information, consisting of 56 continuous morphology parameters, grouped into 4 legs, with 14 parameters per leg: x: the x-coordinate of the hip joint y: the y-coordinate of the hip joint z: the z-coordinate of the hip joint a: the angle of the hip joint b: the angle of the knee joint hip center: the center of the hip joint hip range: the range of the hip joint knee center: the center of the knee joint knee range: the range of the knee joint hip size: the size of the hip joint knee size: the size of the knee joint foot center: the center of the foot joint foot range: the range of the foot joint foot size: the size of the foot joint

879

You are given the following existing designs and their corresponding performance scores:

880
881
882
883
884
885

Robot Morphology Design: [0.11, 0.14, 0.0, 0.46, -2.88, 0.14, -0.34, 0.44, 0.37, 0.79, -1.21, 1.19, 0.09, 0.1, -0.09, 0.11, 0.0, -0.35, 3.76, -0.25, 0.23, 0.4, 0.58, 0.79, -0.31, 1.03, 0.1, 0.1, -0.09, -0.11, 0.0, 0.26, 2.66, 0.18, 0.47, 0.34, 0.45, 0.72, -0.64, 1.04, 0.1, 0.09, 0.11, -0.1, 0.0, -0.51, -2.27, 0.26, -1.03, 0.38, 0.43, 0.76, -0.9, 0.93, 0.09, 0.09], Performance Score: -880.46

886
887
888
889
890

...
Robot Morphology Design: [0.1, 0.13, 0.0, -0.04, -3.55, -0.09, -0.17, 0.42, 0.51, 0.78, -1.24, 0.99, 0.1, 0.1, -0.11, 0.11, 0.0, -0.03, 4.01, 0.2, 0.23, 0.43, 0.73, 0.82, -0.33, 0.98, 0.1, 0.1, -0.09, -0.12, 0.0, 0.06, 3.37, -0.03, 0.36, 0.31, 0.61, 0.61, -0.78, 0.92, 0.1, 0.09, 0.1, -0.09, 0.0, -0.22, -2.74, -0.18, -0.71, 0.42, 0.32, 0.76, -0.43, 0.94, 0.1, 0.09], Performance Score: 199.36

891
892
893

Please propose a new robot morphology design to maximize the performance score. Each feature should be a float number. Each number should be rounded to two decimal places.

894
895
896
897
898
899
900
901
902
903

**In-Context Prompt for TF8**

904
905
906

You are a helpful optimization assistant that will help us generate a new length-8 optimal DNA sequence with maximum binding affinity with a particular transcription factor SIX6 REF R1.

907
908
909

You are given the following existing DNA sequences and their corresponding binding affinities:

910
911

DNA Sequence: ['G', 'G', 'C', 'C', 'G', 'G', 'C', 'C'], Binding Affinity: 0.0

912
913

...
DNA Sequence: ['G', 'T', 'G', 'G', 'G', 'C', 'G', 'A'], Binding Affinity: 0.44

914
915
916
917

Please propose a new DNA sequence that is different from the existing DNA sequences and has higher binding affinity than the existing DNA sequences. The DNA sequences should be in the format of A, C, G, T. The new DNA sequence should be different from the existing DNA sequences in at least 1 position.

Table 3: Experimental results in 50-th percentile normalized scores on four tasks for comparison.

| Method | Ant Morphology | D'Kitty Morphology | TF Bind 8 | TF Bind 10 | Rank Mean | Rank Median |
|---|---|---|---|---|---|---|
| $\mathcal{D}(\textbf{best})$ | 0.565 | 0.884 | 0.439 | 0.467 | – | – |
| Grad-mean | $0.378 \pm 0.006$ | $0.888 \pm 0.001$ | $0.439 \pm 0.000$ | $0.477 \pm 0.000$ | 5.8/16 | 5.0/16 |
| Grad-EI | $0.373 \pm 0.008$ | $0.902 \pm 0.007$ | $\underline{0.451} \pm \underline{0.007}$ | $0.450 \pm 0.005$ | 6.8/16 | 7.0/16 |
| COMs | $0.372 \pm 0.012$ | $0.896 \pm 0.004$ | $0.439 \pm 0.000$ | $0.473 \pm 0.008$ | 6.5/16 | 5.0/16 |
| ICT | $0.384 \pm 0.001$ | $\underline{0.908} \pm \underline{0.001}$ | $0.395 \pm 0.009$ | $0.425 \pm 0.009$ | 8.8/16 | 8.5/16 |
| MATCH-OPT | $0.377 \pm 0.007$ | $0.888 \pm 0.003$ | $0.436 \pm 0.001$ | $0.433 \pm 0.000$ | 9.3/16 | 9.0/16 |
| UniSO-T | $\textbf{0.400} \pm \textbf{0.003}$ | $0.906 \pm 0.004$ | $0.443 \pm 0.005$ | $0.457 \pm 0.006$ | 3.8/16 | 3.0/16 |
| CbAS | $0.351 \pm 0.001$ | $0.893 \pm 0.001$ | $0.390 \pm 0.001$ | $0.431 \pm 0.005$ | 12.8/16 | 13.5/16 |
| ExPT | $\textbf{0.400} \pm \textbf{0.008}$ | $0.903 \pm 0.002$ | $0.350 \pm 0.000$ | $0.464 \pm 0.005$ | 6.8/16 | 5.5/16 |
| MIN | $0.366 \pm 0.001$ | $0.861 \pm 0.004$ | $0.328 \pm 0.008$ | $0.448 \pm 0.009$ | 13.5/16 | 14.0/16 |
| BONET | $0.389 \pm 0.006$ | $0.895 \pm 0.004$ | $0.413 \pm 0.009$ | $0.472 \pm 0.007$ | 6.3/16 | 6.0/16 |
| ORPO | $0.362 \pm 0.003$ | $0.787 \pm 0.004$ | $0.339 \pm 0.005$ | $0.399 \pm 0.003$ | 11.5/16 | 12.5/16 |
| GTG | $0.374 \pm 0.013$ | $0.888 \pm 0.000$ | $0.418 \pm 0.013$ | $0.443 \pm 0.011$ | 9.8/16 | 9.5/16 |
| DDOM | $0.348 \pm 0.008$ | $0.886 \pm 0.001$ | $0.401 \pm 0.003$ | $0.448 \pm 0.006$ | 12.5/16 | 12.0/16 |
| CMA-ES | $0.368 \pm 0.002$ | $0.695 \pm 0.003$ | $0.426 \pm 0.003$ | $0.471 \pm 0.002$ | 10.5/16 | 10.0/16 |
| MCTS-transfer | $0.374 \pm 0.006$ | $0.895 \pm 0.008$ | $0.359 \pm 0.007$ | $\underline{0.478} \pm \underline{0.004}$ | 8.0/16 | 8.0/16 |
| $\textbf{dLLM}_{(\text{ours})}$ | $\underline{0.398} \pm \underline{0.014}$ | $\textbf{0.910} \pm \textbf{0.021}$ | $\textbf{0.463} \pm \textbf{0.015}$ | $\textbf{0.482} \pm \textbf{0.009}$ | **1.5/16** | **1.0/16** |

---

**In-Context Prompt for TF10**

You are a helpful optimization assistant that will help us generate a new length-10 optimal DNA sequence with maximum binding affinity with a particular transcription factor SIX6 REF R1.

You are given the following existing DNA sequences and their corresponding binding affinities:

DNA Sequence: ['T', 'C', 'C', 'A', 'C', 'G', 'A', 'A', 'G', 'A'], Binding Affinity: -1.86
...
DNA Sequence: ['G', 'C', 'T', 'T', 'G', 'G', 'A', 'A', 'C', 'A'], Binding Affinity: 0.01

Please propose a new DNA sequence that is different from the existing DNA sequences and has higher binding affinity than the existing DNA sequences. The DNA sequences should be in the format of A, C, G, T. The new DNA sequence should be different from the existing DNA sequences in at least 1 position.

---

## A.3 MEDIAN RESULTS

We further report the 50-th percentile normalized scores in Table 3, where best and second-best results are **bolded** and underlined, respectively. Our method *dLLM* achieves the best overall mean and median rank across 16 methods, ranking first on D'Kitty, TF8, and TF10, and third on Ant. This demonstrates that *dLLM* consistently maintains strong performance.

## A.4 DIFFERENT TASK DESCRIPTIONS

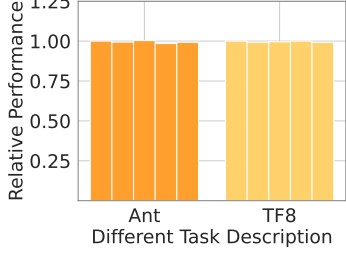

Figure 6: Relative performance under different task descriptions.

We evaluate dLLM's robustness to in-context prompt variation by replacing the default task description with five GPT-5-generated templates. As shown in Figure 6, normalized scores remain stable—ranging from $0.642$ to $0.654$ on Ant and $0.869$ to $0.876$ on TF8—demonstrating robustness to stylistic changes in task descriptions.

