# OpenReview forum: "Diffusion Large Language Models for Black-Box Optimization"
_ICLR.cc/2026/Conference — Submitted to ICLR 2026_

### Official Review · Reviewer_mECb · 2025-10-16

**Soundness:** 2
**Presentation:** 3
**Contribution:** 2
**Rating:** 4
**Confidence:** 5

**Summary:**

This manuscript looks at the utility of diffusion LLMs for solving black-box optimization problems – specifically evaluating their method against baseline optimizations strategies on 4 tasks from Design Bench. The authors combine diffusion LLMs with a tree search strategy where the goal of the diffusion LLM is to navigate a tree of possible designs. The proposed methods ranks first across the 4 evaluated tasks when compared against baseline methods.

**Strengths:**

- This is an interesting and intuitive method. I have personally been working on offline MBO and Design Bench for a while now, and think that using diffusion LLMs is a natural strategy for offline MBO problems. The strategy to also incorporate MCTS is also interesting.
- This manuscript is well-written. I found the language easy to understand and follow, with minimal (if any) grammatical issues that would preclude scientific understanding.

**Weaknesses:**

My main concerns with this work relate to the empirical evaluation of the proposed method. I detail my comments point-by-point below:
1. The performance of the OPRO baseline helps illustrate the limitation of traditional autoregressive LLMs in capturing potential right-to-left dependencies in designs, as noted by the authors. However, several methods could have been used to significantly boost the performance of the baseline – for example, chain-of-thought prompting (encouraging the model to reason about the specific task/domain before answering); or providing additional domain-specific knowledge (e.g., context about the SIX6 REF R1 transcription factor); or even using a more performant LLM (maintaining the same size as the diffusion LLM is fine).
2. I do not understand why the sizes of the offline dataset were kept so small. Ablating the offline dataset size between 2 and 20 is helpful, but in many tasks (and also in Design Bench), the sizes of the offline datasets can be in the thousands, which is orders of magnitude larger than what is reported here. Because many of the baselines are data driven, I would be surprised to see any of the baselines perform particularly well here. At the very least, the authors should sweep over much larger ranges of the size of the offline dataset. In particular, it would be helpful to know at what point do existing methods surpass the proposed dLLM method (if ever).
3. Somewhat of a continuation of comment (2) above, but it doesn’t make sense that the offline dataset would be so small (implying that oracle evaluations are very expensive), and yet the number of final designs that you can evaluate with the oracle at the very end is so much larger (128). The reasoning would be that if the oracle is truly inexpensive enough to allow for 128 final evaluations in real life, then we should have spent time building a better dataset first to have more data to learn from. To end, it would be important to ablate the number oracle evaluation budget – particularly focusing on extremely small values (e.g., 1 instead of 128 allowed evaluations).
4. Similar to 2, a naïve strategy to improve the performance of both dLLM and baselines is to use the best 10 (or however many) designs in the offline dataset, rather than picking a random sample of 10 designs. It would also be helpful to see how it performs if sampling the worst N designs as well (although not as important).
5. It would also be helpful to see an ablation study on $J$ in estimating the node’s reward in line 260.
6. Similarly, ablating $\omega$ in (4) is important to characterize the impact of the trade-off between exploration and exploitation on algorithm performance.
7. Similarly, ablation the temperature parameter of the diffusion LLM is also important, as I would imagine it would similarly impact the trade-off between exploration and exploitation.
8. TFBind8 and TFBind10 are very similar tasks, and overall it feels like the number of baseline tasks evaluated are relatively small. The manuscript would benefit from evaluating the performance of the proposed method on NAS from the original Design-Bench paper, and/or on other benchmarking tasks (e.g., the Warfarin and LogP tasks from [this paper](https://arxiv.org/abs/2402.06532)). This would help better characterize the generalizability of the method across a more representative set of tasks.

Minor Comments:
- Line 108: It would be helpful to include a brief overview of what the UCT score is (e.g., what the acronym stands for) and the main intuition behind it. I understand that it’s formally defined in Eq (4), although it’s first introduced earlier in the paper.

**Questions:**

9. In Line 210, it is mentioned that "any additional invalid designs… are excluded." How often does this actually happen? It would be helpful to report to know how much of the compute resources are being used to generate valid vs invalid designs.
10. I am not sure if I fully understand the claimed "left-to-right" limitation of traditional autoregressive LLMs. For the full designs that are included as context in the LLM prompt, attention should capture dependencies bidirectionally between the tokenized components of any given design.
11. I might have missed this, but I’m not sure how many seeds/experimental runs were used to report the experimental results in Table 1. Also, are the error bars Standard Deviation, SEM, or something else?

---

> ### Author Response · Authors · 2025-11-22
>
> ## Overall
> Dear Reviewer mECb,
>
> We sincerely appreciate your thorough and constructive review. Your insights have guided substantial improvements, and our detailed replies and revisions are presented below.
>
> ## Weakness
> > The performance of the OPRO baseline helps illustrate the limitation of traditional autoregressive LLMs in capturing potential right-to-left dependencies in designs, as noted by the authors. However, several methods could have been used to significantly boost the performance of the baseline – for example, chain-of-thought prompting (encouraging the model to reason about the specific task/domain before answering); or providing additional domain-specific knowledge (e.g., context about the SIX6 REF R1 transcription factor); or even using a more performant LLM (maintaining the same size as the diffusion LLM is fine).
>
> We thank the reviewer for the helpful suggestions. We plan to test stronger OPRO variants—e.g., CoT prompting, domain-specific context, and more capable autoregressive LLMs. We plan to include this as part of **future work**. We also note that such enhancements are beyond the scope of this study, as similar techniques could likewise improve diffusion LLMs; our goal here is to compare prototype AR LLMs with dLLM.
>
> > I do not understand why the sizes of the offline dataset were kept so small. Ablating the offline dataset size between 2 and 20 is helpful, but in many tasks (and also in Design Bench), the sizes of the offline datasets can be in the thousands, which is orders of magnitude larger than what is reported here. Because many of the baselines are data driven, I would be surprised to see any of the baselines perform particularly well here. At the very least, the authors should sweep over much larger ranges of the size of the offline dataset. In particular, it would be helpful to know at what point do existing methods surpass the proposed dLLM method (if ever).
>
> > Somewhat of a continuation of comment (2) above, but it doesn’t make sense that the offline dataset would be so small (implying that oracle evaluations are very expensive), and yet the number of final designs that you can evaluate with the oracle at the very end is so much larger (128). The reasoning would be that if the oracle is truly inexpensive enough to allow for 128 final evaluations in real life, then we should have spent time building a better dataset first to have more data to learn from. To end, it would be important to ablate the number oracle evaluation budget – particularly focusing on extremely small values (e.g., 1 instead of 128 allowed evaluations).
>
> We appreciate the reviewer’s suggestion. We plan to extend our ablation to include substantially larger offline dataset sizes (e.g., 500 and 1000 samples) to study how dLLM and baselines behave as more data becomes available. We plan to include this as part of **future work**.
>
> > Similar to 2, a naïve strategy to improve the performance of both dLLM and baselines is to use the best 10 (or however many) designs in the offline dataset, rather than picking a random sample of 10 designs. It would also be helpful to see how it performs if sampling the worst N designs as well (although not as important).
>
> We would like to clarify that the 10 examples in our few-shot setting are uniformly sampled from the offline dataset rather than chosen arbitrarily. This ensures representative coverage across the score distribution rather than favoring either high- or low-performing regions.
>
> In addition, we are currently extending our experiments along the reviewer’s suggestions. Specifically, we plan to evaluate (i) different oracle-evaluation budgets, including extremely small budgets (e.g., 1 instead of 128), and (ii) offline datasets constructed from the best or worst $N$ designs to study how the quality of initial examples affects both dLLM and all baselines. We plan to include this as part of **future work**.
>
> > It would also be helpful to see an ablation study on $J$ in estimating the node’s reward in line 260.
>
> We acknowledge the importance of validating the sensitivity of $J$. We plan to conduct additional experiments sweeping $J \in \\{1, 3, 5, 7, 9\\}$ to evaluate how varying this parameter affects the node's reward estimation. We plan to include this as part of **future work**.
>
> > Similarly, ablating $w$ in (4) is important to characterize the impact of the trade-off between exploration and exploitation on algorithm performance.
>
> To better characterize the trade-off between exploration and exploitation, we plan to run ablation studies on $w$, specifically testing $w \in \\{0.5, 2\\}$ alongside our default settings. We plan to include this as part of **future work**.

---

> ### Author Response · Authors · 2025-11-22
>
> > Similarly, ablation the temperature parameter of the diffusion LLM is also important, as I would imagine it would similarly impact the trade-off between exploration and exploitation.
>
> While our initial experiments employed a default temperature of $1.0$, we agree that analyzing this parameter is crucial. We are currently running sensitivity tests with different temperature values to assess the diffusion LLM's exploration behavior. We plan to include this as part of **future work**.
>
> > TFBind8 and TFBind10 are very similar tasks, and overall it feels like the number of baseline tasks evaluated are relatively small. The manuscript would benefit from evaluating the performance of the proposed method on NAS from the original Design-Bench paper, and/or on other benchmarking tasks (e.g., the Warfarin and LogP tasks from this paper). This would help better characterize the generalizability of the method across a more representative set of tasks.
>
> We appreciate the reviewer's suggestion to broaden the evaluation suite. Regarding the NAS task, we found it to be prohibitively computationally intensive to complete. However, to address the concern regarding task diversity and generalizability, we are currently conducting experiments on the other suggested benchmarks. We plan to include this as part of **future work**.
>
> > Line 108: It would be helpful to include a brief overview of what the UCT score is (e.g., what the acronym stands for) and the main intuition behind it. I understand that it’s formally defined in Eq (4), although it’s first introduced earlier in the paper.
>
> We have revised the text to explicitly expand the acronym Upper Confidence Bound for Trees (UCT) and added a clause explaining its core mechanism: "which balances the exploitation of high-value nodes with the exploration of less-visited ones."
>
> ## Questions
> > In Line 210, it is mentioned that "any additional invalid designs… are excluded." How often does this actually happen? It would be helpful to report to know how much of the compute resources are being used to generate valid vs invalid designs.
>
> As we emphasized in line 209-210, we restricted sampling to valid logits only. Therefore, for discrete DNA sequence design tasks, the generated designs are always valid. For continuous designs, the valid ratio is also fairly high, with $87$\% valid ratio across the two continuous tasks.
>
> > I am not sure if I fully understand the claimed "left-to-right" limitation of traditional autoregressive LLMs. For the full designs that are included as context in the LLM prompt, attention should capture dependencies bidirectionally between the tokenized components of any given design.
>
> Thanks for the comment. We clarify that we focus on the generation process of each model. When generating tokens from left to right, the autoregressive model is unable to attend to the right, as the future tokens are not generated yet. However, for the diffusion language model, since its generation order is not strictly left-to-right, the model is able to capture the bidirectional dependencies during the generation process.
>
> For example, in DNA sequence design, the selection of a specific unit should ideally be informed by both its prefix and its suffix. While AR models are restricted to conditioning on the prefix, diffusion LMs capture dependencies from both directions.
>
> > I might have missed this, but I’m not sure how many seeds/experimental runs were used to report the experimental results in Table 1. Also, are the error bars Standard Deviation, SEM, or something else?
>
> All experiments are conducted with $8$ different seeds, and the error bars are the standard deviations.

---

> > ### Comment · Reviewer_mECb · 2025-11-26
> >
> > I thank the authors for their detailed comments and appreciate that they were able to address some of my questions. I look forward to reviewing the updated experimental results as they become available. I have no additional concerns to raise at this time.
> >
> > As we head into the Thanksgiving holiday (in the United States), I also wanted to let the authors know that I will be active and online - at least for me personally, please feel free to continue the discussion over the coming days and I will do my best to respond in a timely manner. (Of course, it is totally fine to take the holiday off as well! No worries either way)

---

### Official Review · Reviewer_78bn · 2025-10-18

**Soundness:** 2
**Presentation:** 3
**Contribution:** 2
**Rating:** 2
**Confidence:** 4

**Summary:**

The paper proposes a new method, dLLM, for offline black-box optimization (BBO) in few-shot settings. The core idea is to leverage pre-trained diffusion-based large language models, which are hypothesized to be better suited for design tasks with bidirectional dependencies than their autoregressive counterparts. The method consists of two main components: (1) an "in-context denoising" module, which prompts the diffusion LLM with a task description and an offline dataset to iteratively refine a masked design, and (2) a "masked diffusion tree search" (MDTS), which frames the denoising process as a Monte Carlo Tree Search (MCTS). The search is guided by rewards derived from the Expected Improvement (EI) of a Gaussian Process (GP) trained on the offline data, effectively balancing exploration and exploitation. The authors demonstrate that dLLM achieves state-of-the-art performance on four tasks from the design-bench benchmark.

**Strengths:**

- The proposed method consists of a straightforward integration of several existing techniques. The use of a GP's EI as a reward signal for an MCTS-guided denoising process is intuitive
- The paper is overall well-written and the ablation studies are well-designed to demonstrate the importance of each of the proposed components
- This work highlights a promising new direction for LLM-based optimizers by moving from autoregressive to diffusion-based models. If the efficiency concerns can be addressed, this approach could have a profound impact on scientific and engineering design problems

**Weaknesses:**

- The paper's most significant weakness is its lack of theoretical analysis. Specifically, the proposed MDTS is presented without any formal guarantees (such as regret analysis), despite being built upon the MCTS framework
- The method's guidance relies on EI from a GP trained on only a few examples in very high dimensional spaces. However, it is well-known that GPs with standard kernels often perform poorly in such settings (curse of dimensionality), and their uncertainty estimates can be unreliable [1]-[4]. The paper does not analyze the quality of the GP fit or the sensitivity of the method to the GP's configuration (e.g., kernel choice), making it unclear how robust the approach is.
- The paper also lacks any discussion on the computational complexity/cost of the proposed method. This omission makes it difficult to assess the practical viability of the method

[1] D. Eriksson and M. Jankowiak, "High-dimensional Bayesian optimization with sparse axis-aligned subspaces," UAI, 2021.

[2] R. C. Suwandi et al., "Adaptive kernel design for Bayesian optimization is a piece of CAKE with LLMs," arXiv preprint arXiv:2509.17998, 2025.

[3] Z. Wang et al., "Bayesian optimization in a billion dimensions via random embeddings," Journal of Artificial Intelligence Research, 2016.

[4] M. Riccardo et al., "High-dimensional Bayesian optimization using low-dimensional feature spaces," Machine Learning, 2020.

**Questions:**

- How are the few-shot examples in the offline dataset ordered within the in-context prompt? Have the authors tested the sensitivity of dLLM's performance to the permutation of these examples, as this is a known factor that can significantly impact the performance of in-context learning
- The introduction and Section 4.4 argue that diffusion models excel due to "bidirectional modeling", which is a central motivation for this work. Could the authors provide a concrete example from the tasks that illustrates this? For instance, can the authors show a case where an autoregressive model makes a greedy left-to-right decision that proves suboptimal, whereas dLLM's iterative refinement corrects this by considering both left and right context simultaneously?
-  In the UCT score, how does the model likelihood term $p_\theta(x_{s,i}|x_t)$ interact with the value estimate $V(x_{s,i})$ during the search? Is there a risk that the LLM's prior could overwhelm the optimization signal from the GP, especially early in the search when $V$ is poorly estimated?
- Lines 205-208: "we then remask the least confident tokens." How is token "confidence" formally defined and calculated in this context?
- Did you experiment with different kernels or alternative acquisition functions?
- How does the performance of dLLM change if a simpler reward, like the GP's predictive mean, is used instead of EI?

---

> ### Author Response · Authors · 2025-11-22
>
> ## Overall
>
> Dear Reviewer 78bn,
>
> We are grateful for your comprehensive comments. We provide point-by-point responses and corresponding updates below.
>
> ## Weakness
>
> > The paper's most significant weakness is its lack of theoretical analysis. Specifically, the proposed MDTS is presented without any formal guarantees (such as regret analysis), despite being built upon the MCTS framework
>
> We appreciate the reviewer’s suggestion to provide a more formal analysis of our masked diffusion tree search (MDTS). Our design of MDTS is deliberately grounded in the standard UCT framework (Eq. (4)), whose convergence and regret properties under idealized assumptions (finite action spaces, stationary transitions, unbiased rewards) have been well studied in the MCTS literature. In our setting, however, the search operates over high-dimensional design spaces with (i) stochastic proposals produced by a diffusion LLM and (ii) rewards computed via a GP-based EI acquisition function. Deriving regret guarantees that explicitly account for both the generative model and the surrogate uncertainty would require strong additional assumptions on the diffusion LLM and the GP, and constitutes a substantial theoretical effort that is orthogonal to the empirical focus of this work.
>
> Importantly, this situation is completely aligned with many successful modern MCTS-based systems that integrate tree search with powerful neural generative or value models. Even in prominent examples such as neural-network–guided MCTS for large board games (e.g., AlphaGo/AlphaZero), the full pipeline does not admit end-to-end regret or convergence guarantees—the theoretical results apply only to the underlying UCT family under idealized conditions. Consequently, prior works have primarily relied on empirical validation rather than complete theoretical characterization when powerful function approximators are involved.
>
> Following this line, our work emphasizes the algorithmic design and empirical behavior of MDTS. We provide extensive evidence of its effectiveness and robustness: (1) ablations removing MDTS or replacing the diffusion LLM with a task-specific diffusion model (Table 2), both of which lead to clear performance drops; (2) sensitivity analyses over tree depth, branching factor, and offline dataset size (Figs. 3–5); and (3) robustness to different task descriptions and diffusion LLM backbones (Appendix A.4). These results consistently demonstrate that MDTS is crucial for dLLM and behaves stably across diverse configurations.
>
> We have added a brief discussion in the revision highlighted in read to clarify that MDTS is proposed as a practical UCT-based search heuristic, and that developing formal regret guarantees for diffusion-LLM–guided MCTS in offline BBO is an interesting direction for future theoretical work.
>
> > The method's guidance relies on EI from a GP trained on only a few examples in very high dimensional spaces. However, it is well-known that GPs with standard kernels often perform poorly in such settings (curse of dimensionality), and their uncertainty estimates can be unreliable [1]-[4]. The paper does not analyze the quality of the GP fit or the sensitivity of the method to the GP's configuration (e.g., kernel choice), making it unclear how robust the approach is.
>
>
> We thank the reviewer for the insightful comment. We agree that GPs with standard kernels may be unreliable in high-dimensional spaces. In our framework, however, the GP is not the main optimization driver but serves as a lightweight reliability filter to rank candidates generated by the diffusion LLM. The primary optimization signal comes from the diffusion LLM’s generative prior, while the GP’s EI score is used only to prune clearly suboptimal designs. This design substantially reduces the impact of potential GP miscalibration. We plan to include this as part of **future work**.
>
> > The paper also lacks any discussion on the computational complexity/cost of the proposed method. This omission makes it difficult to assess the practical viability of the method
>
> We would like to clarify that the **runtime and computational cost** of our method are already reported in the paper. As stated in **Section 4.3 (lines 349–350)**, *“All experiments are conducted on a single NVIDIA A100 GPU with 80 GB of memory. In terms of runtime, a full search episode takes approximately 30 minutes on Ant and 40 minutes on TF8.”*

---

> ### Author Response · Authors · 2025-11-22
>
> ## Questions
> > How are the few-shot examples in the offline dataset ordered within the in-context prompt? Have the authors tested the sensitivity of dLLM's performance to the permutation of these examples, as this is a known factor that can significantly impact the performance of in-context learning
>
> We acknowledge that the ordering of few-shot examples in the in-context prompt may influence performance. We plan to conduct a systematic permutation study to evaluate this sensitivity.
>
> > The introduction and Section 4.4 argue that diffusion models excel due to "bidirectional modeling", which is a central motivation for this work. Could the authors provide a concrete example from the tasks that illustrates this? For instance, can the authors show a case where an autoregressive model makes a greedy left-to-right decision that proves suboptimal, whereas dLLM's iterative refinement corrects this by considering both left and right context simultaneously?
>
> In our experiments, the **OPRO** baseline serves exactly as the **autoregressive left-to-right model** for comparison. As shown in **Table 1**, OPRO performs notably worse than our diffusion LLM (dLLM) across all tasks.
> This performance gap empirically demonstrates the importance of **bidirectional modeling**: unlike OPRO’s greedy left-to-right generation, dLLM refines candidates iteratively and jointly conditions on both left and right context at every denoising step, which helps correct suboptimal early decisions and capture global dependencies within sequences.
>
> While providing an exact per-token failure case is challenging, our DNA-sequence design example highlights why bidirectional reasoning is essential: the binding affinity of a nucleotide depends on both its prefix and suffix, making unidirectional generation fundamentally less expressive for such tasks.
>
> > In the UCT score, how does the model likelihood term $p_\theta(x_{s,i}|x_t)$ interact with the value estimate $V(x_{s,i})$ during the search? Is there a risk that the LLM's prior could overwhelm the optimization signal from the GP, especially early in the search when $V$ is poorly estimated?
>
> In the UCT score, the likelihood term $p_\theta(x_{s,i}\mid x_t)$ and the value estimate $V(x_{s,i})$ play complementary roles. The value term provides the exploitation signal through the GP-based Expected Improvement, while the likelihood term serves as a prior within the exploration component, encouraging the search to first explore denoising trajectories that are more plausible under the diffusion LLM.
>
> Importantly, this prior cannot overwhelm the optimization signal. Early in the search, all nodes have very small visit counts, so the exploration term is large and the LLM prior only influences *where* initial exploration goes. As nodes are evaluated and EI rewards are back-propagated, their visit counts grow and the exploration term decays as $\mathcal{O}(1/\sqrt{N})$, causing the UCT score to become increasingly dominated by $V(x_{s,i})$. The final designs are selected solely based on their EI scores, ensuring that the GP---not the LLM prior---ultimately determines which candidates are preferred.
>
> > Lines 205-208: "we then remask the least confident tokens." How is token "confidence" formally defined and calculated in this context?
>
> We follow Nie et al. (2025) [1], which adopts the confidence formulation introduced in MaskGIT (Chang et al., 2022) [2]. Specifically, for each masked location $i$, the diffusion LLM samples a token $y_i^{(t)}$ from its predicted categorical distribution $p_i^{(t)} \in \mathbb{R}^K$. The corresponding **confidence score** is the model’s predicted probability of the sampled token.
>
> [1] Nie et al., *Large Language Diffusion Models*, 2025.
> [2] Chang et al., *MaskGIT: Masked Generative Image Transformer*, CVPR 2022.
>
> > Did you experiment with different kernels or alternative acquisition functions?
>
> We appreciate the suggestion. We plan to experiment with alternative GP kernels (e.g., Matérn-5/2) and additional acquisition functions such as UCB. We plan to include this as part of **future work**.
>
>
> > How does the performance of dLLM change if a simpler reward, like the GP's predictive mean, is used instead of EI?
>
> We chose EI as the reward because it leverages both the GP predictive mean and its uncertainty, which is particularly important in few-shot BBO. Nonetheless, we plan to run experiments using the GP predictive mean as a simpler alternative to assess its impact, including this as part of **future work**.

---

> ### Comment · Reviewer_78bn · 2025-11-26
> **Reply to Authors' Rebuttal**
>
> Thank you for the detailed rebuttal. I appreciate the authors' efforts, and several of my original concerns have been addressed. However, after reading the responses, there are a few points I would like clarified:
>
> - How much slower is dLLM exactly compared to strong GP or COMs baselines on the same tasks and hardware? A 30–40 minute runtime on an A100 suggests a significant slowdown, which may impact practicality, especially if not justified by substantial performance gains.
>
> - If the GP/EI is only a "lightweight reliability filter," then why build an expensive MCTS search on top of it? This undermines the core motivation. Can you show that using just the GP mean (rather than EI) does not collapse performance? Otherwise, the importance of uncertainty-aware guidance remains unsubstantiated.
>
> - I still want to see results on kernel sensitivity, acquisition function comparison, and especially the ablation between EI and predictive mean. I am willing to reassess my score if these results are included during the rebuttal.
>
> For now, I decided to maintain my current recommendation.

---

### Official Review · Reviewer_wStR · 2025-10-25

**Soundness:** 3
**Presentation:** 2
**Contribution:** 3
**Rating:** 4
**Confidence:** 4

**Summary:**

This work considers black box optimization using the in context abilities of LLMs. But autoregressive LLMs can be poor planners, when bi-directional dependencies exists. Therefore, this work considers diffusion language models for this task. The paper introduces Masked diffusion tree search to do black box optimization with some in-context information, which is a UCT based MCTS algorithm. The algorithm's performance is then evaluated on various BBO tasks to show significant gains.

**Strengths:**

The empirical gains are impressive and showcases a strong case for pretrained diffusion models being used for BBO in planning problems. In this case, an autoregressive LLM base methods  such as ORPO do not even perform as well as the classical Gaussian process based methods. The algorithm design is a simple adaptation of MCTS to this case. There are extensive ablations on the effect of tree depth, branching factor and offline dataset size.

**Weaknesses:**

The paper is poorly written on a technical level and the algorithmic and experimental details have not been explain within the paper. See "Questions" for some specific queries in this regard. Given these drawbacks, I cannot recommend acceptance for this work.

Since this paper uses a large diffusion model whereas the previous works in this domain use simple techniques such as Gaussian process, a comparison of computation complexities of various methods is important. This is not provided in the paper

**Minor**
* Some additional references to consider:
https://arxiv.org/abs/2410.07432 considers reasoning with transformers using COT inspired algorithms to solve planning problems. The benefits of solving bi-directional planning problems using diffusion models have been explored in the literature (see https://arxiv.org/pdf/2410.14157,  https://arxiv.org/pdf/2502.13450 and references therein). This is not exactly BBO, but can be relevant.

* The authors use the phrase “domains such as DNA” – which seems to be confusing.

**Questions:**

In algorithm 1, what is t? The selection procedure here tells us to pick $x_t$ based on $x_1$ but the line 234 tells us how to pick $x_s$ based on $x_t$. There is some notation overload causing confusion here.

What is the value function V(.) here and how is it calculated? (for instance, in the DNA affinity task). Where is this initialized in the algorithm? Is this tabular? In this case even with length 8, the number of sequences would be ~65k.

In line 234 **selection** –  what does “select child based on UCT score” mean? Does this mean you are picking the child with the maximum UCT score?

In **Expansion**: Does this mean that we generate $x_{0,i}$ which is the completion and then re-mask it randomly to generate new leaves $x_{s,i}$ ? In this case shouldn't the remask ratio be s instead of $s/t$ ?

In **Evaluation**: I did not understand how the predictive mean and variance are obtained through the Gaussian process given that ATGC are discrete categories in the DNA affinity case. Can you please point me to references where such modeling is done? Also, the details about the updation of this gaussian process is not available. Is it the case that only the offline data points have affinity score available and a GP is used to “fit” the data to other generated points? I understand this is so given the algorithm pseudocode. I would also appreciate a reference for the expression for “Expected Improvement” and its meaning.

---

> ### Author Response · Authors · 2025-11-22
>
> ## Overall
>
> Dear Reviewer wStR,
>
> Thank you for the thoughtful and detailed feedback. Your comments have significantly helped us refine the manuscript, and we address each point carefully in the responses below.
>
> ## Weakness
> > Since this paper uses a large diffusion model whereas the previous works in this domain use simple techniques such as Gaussian process, a comparison of computation complexities of various methods is important. This is not provided in the paper
>
> We agree that including computational cost comparisons would strengthen the paper. We plan to include this as part of **future work**. We would also like to clarify that Section 4.3 (lines 349–350) already reports the runtime of our approach: a full masked-diffusion tree search requires approximately 30 minutes on Ant and 40 minutes on TF8 using a single NVIDIA A100 GPU. As expected, GP models are generally more computationally efficient.
>
> However, as highlighted in prior work—e.g., Bidirectional Learning for Offline Infinite-width Model-based Optimization (NeurIPS 2022, Appendix A.3)—in many real-world offline black-box optimization scenarios such as biological or chemical design, the dominant cost lies in evaluating the unknown objective function rather than in running the optimization algorithm itself. Consequently, moderate differences in algorithmic runtime have limited impact in production settings, where solution quality is typically prioritized over compute cost.
>
> > Some additional references to consider: https://arxiv.org/abs/2410.07432 considers reasoning with transformers using COT inspired algorithms to solve planning problems. The benefits of solving bi-directional planning problems using diffusion models have been explored in the literature (see https://arxiv.org/pdf/2410.14157, https://arxiv.org/pdf/2502.13450 and references therein). This is not exactly BBO, but can be relevant.
>
> We thank the reviewer for the helpful suggestions. We appreciate the pointers to recent works on reasoning and bidirectional planning with diffusion models. We agree that these studies are relevant and complementary to our approach, and we included them in the Related Work section of the revised version.
>
> > The authors use the phrase “domains such as DNA” – which seems to be confusing.
>
> We agree that the phrase “domains such as DNA” may be ambiguous. We have revised it to “DNA sequence design” in the revised version for greater clarity.
>
> ## Questions
> > In algorithm 1, what is t? The selection procedure here tells us to pick $x_t$ based on $x_1$ but the line 234 tells us how to pick $x_s$ based on $x_t$. There is some notation overload causing confusion here.
>
> We thank the reviewer for catching this notation ambiguity. In Algorithm 1, the selection process indeed traverses the tree from the root node $x_1$ to a selected node $x_t$. In line 234, the description refers to the selection from a parent node to its child node $x_{s,i}$. We acknowledge this minor overload of notation and revised the notation in line 234 to clarify it by changing the parent node symbol from $x_t$ to $x_\tau$ in the revised version.
>
> > What is the value function V(.) here and how is it calculated? (for instance, in the DNA affinity task). Where is this initialized in the algorithm? Is this tabular? In this case even with length 8, the number of sequences would be $\approx$65k.
>
> The value function $V(\cdot)$ in our method is a node-specific statistic maintained by the Monte Carlo Tree Search (MCTS). It is not a learned value model, nor is it stored as a full tabular function over the entire design space. For each newly expanded node $\mathbf{x}_t$, we initialize
> $V(\mathbf{x}_t)=0$, and we update it only when the node is visited, following the backpropagation rule provided in Eq.~(7) of the paper.
>
> Importantly, $V(\cdot)$ is defined only for nodes actually explored by the search tree. For example, in the DNA affinity task, although the full sequence space contains $4^8 \approx 65{,}536$ sequences, MCTS expands only a small subset of them, so no full tabular representation is required. Thus, the value function is lightweight, local to the search tree, and updated incrementally during MCTS.

---

> ### Author Response · Authors · 2025-11-22
>
> > In line 234 selection – what does “select child based on UCT score” mean? Does this mean you are picking the child with the maximum UCT score?
>
> Yes. During selection, we start from the root node and iteratively choose the child with the maximum UCT score, treating it as the new parent node, until we reach a node with unexpanded actions where expansion is performed, following the standard Monte Carlo Tree Search procedure.
>
> > In Expansion: Does this mean that we generate $x_{0, i}$ which is the completion and then re-mask it randomly to generate new leaves $x_{s,i}$? In this case shouldn't the remask ratio be s instead of $s/t$?
>
> We thank the reviewer for the question. The remask ratio $s/t$ is defined relative to the previous step $x_t$ rather than the initial fully masked state at $t=1$. At each expansion step, we begin from $x_t$ (which has a mask ratio of $t$), generate its completion $x_0$, and then construct $x_s$ by re-masking a fraction $s/t$ of the tokens that were masked in $x_t$.
>
> > In Evaluation: I did not understand how the predictive mean and variance are obtained through the Gaussian process given that ATGC are discrete categories in the DNA affinity case. Can you please point me to references where such modeling is done? Also, the details about the updation of this gaussian process is not available. Is it the case that only the offline data points have affinity score available and a GP is used to “fit” the data to other generated points? I understand this is so given the algorithm pseudocode. I would also appreciate a reference for the expression for “Expected Improvement” and its meaning.
>
> We thank the reviewer for the question. For discrete design-space tasks such as DNA affinity optimization, we follow Trabucco et al. (ICML 2022) and **map categorical tokens (A/C/G/T)** to **real-valued logits** of a categorical distribution, allowing the Gaussian Process (GP) to operate in a continuous space.
>
> Yes, the GP is **fit only once** on the offline dataset using the available affinity scores and is **not updated** during the optimization process. It serves as a fixed surrogate to estimate the predictive mean and variance for newly generated candidates.
> As already described in Eq.(5), the **Expected Improvement (EI)** criterion used in our work follows the standard definition from Bayesian optimization:
> $$
> \text{EI}(x) = (\mu(x) - f_{\text{best}}) \Phi(z) + \sigma(x) \phi(z),
> \quad \text{where } z = \frac{\mu(x) - f_{\text{best}}}{\sigma(x)}.
> $$
> Here, $\mu(x)$ and $\sigma(x)$ are the GP predictive mean and standard deviation, and $\Phi$, $\phi$ are the CDF and PDF of the standard normal distribution, respectively.
>
> [1] Trabucco et al., *Design-Bench: Benchmarks for Data-Driven Offline Model-Based Optimization*, ICML 2022.

---

### Official Review · Reviewer_9Unr · 2025-10-29

**Soundness:** 2
**Presentation:** 3
**Contribution:** 2
**Rating:** 2
**Confidence:** 3

**Summary:**

This paper proposes dLLM, a new method for offline black-box optimization (BBO) that utilizes pre-trained diffusion large language models. BBO is a classical environment for data-rare scenarios, such as AI for Science, where only a small offline dataset of designs and their scores is available. The authors identify that standard autoregressive LLMs struggle with bidirectional dependencies common in design tasks (e.g., DNA sequences). This work studies a diffusion language model, where the denoising is autoregressive, conditioned on task description, the offline dataset, and an instruction (all concatenated to a natural language prompt). MCTS is also included to balance exploration and exploitation in the denoising process.

**Strengths:**

1. First of all this work has a good presentation. Its way of introducing methodology is easy to follow. The paper also provides a thorough review of related work in Section 5, which, to my understanding, is important because, since 2023, there has been a line of works applying diffusion models/LLMs for black-box optimization. Diffusion LMs are also a part of LLMs, bearing huge similarities and relevantness; it is extremely important to situate this work with respect to the prior works.



2. Related to 1. The selection of baselines is also thorough. The authors compare their method against 15 baselines, including proxy-based methods and various generative models. This includes the must-have prior works in both LLM for BBO (such as OPRO) and diffusion for BBO (such as DDOM).

**Weaknesses:**

1. First, this work extends LLMs for BBO to diffusion LMs. The reason for such extension, is described by: using diffusion models to capture bidirectional dependencies. This does not seem to be wrong, but it is also highly untrivial to evaluate. Have the authors come up with certain mechanism theories to formally describe this to this argument?

2. The originality of the paper is somewhat limited. As mentioned, the work feels like a natural and incremental extension of existing ideas. The field already has "LLMs for BBO" and "diffusion models for BBO". This paper's "in-context denoising" component is a straightforward combination of these two, which has been introduced by Tang et al. 2025 to optimize molecules.


Similarly, the use of MCTS to guide diffusion models is also an existing area of research, which the authors acknowledge, e.g., Tang et al. (2025, which proposes an MCTS framework built on top of diffusion language models for molecular optimization, more specifically, SMILES. This work, on the other hand, has studied DNA sequences, which appear to be even more approachable than SMILES as SMILES encodes complicated structural information and many atoms.

3. The ablation study in Table 2 is a major concern. Removing MDTS (the tree search) leads to a massive performance drop (e.g., 0.876 to 0.798 on TF8; 0.642 to 0.503 on TF10). This strongly suggests that the search framework (MCTS guided by the GP's EI) is doing almost all the work. The dLLM is just acting as a proposal mechanism within this search.

**Questions:**

1. see above

2. To truly isolate the contribution of the dLLM as a better mechanism then AR, could the authors add an ablation study? Specifically, please compare the full dLLM method against a baseline that uses the exact same MDTS framework (same tree depth, branches, GP-EI reward) but replaces the dLLM with a simpler mechanism (e.g., simpler generative models such as a VAE)

3. As mentioned, dLLMs are claimed to be better about "bidirectional dependencies" than LLMS, but no theoretical explanations are given. If we only examine experimental validations, results on TF8/TF10 (DNA sequences) seem to support this, as the gaps to LLMs are largest there. However, it seems performance gaps on the continuous tasks are tiny. Is the method's advantage primarily for discrete, sequence-like design spaces, and less so for continuous ones? Is there a theory to explain this?

---

> ### Author Response · Authors · 2025-11-22
>
> ## Overall
>
> Dear Reviewer 9Unr,
>
> We greatly appreciate your detailed comments, which have been instrumental in improving clarity and quality. Our point-by-point responses and revisions are provided below.
>
> ## Weakness
>
> > First, this work extends LLMs for BBO to diffusion LMs. The reason for such extension, is described by: using diffusion models to capture bidirectional dependencies. This does not seem to be wrong, but it is also highly untrivial to evaluate. Have the authors come up with certain mechanism theories to formally describe this to this argument?
>
> We appreciate the reviewer’s comment regarding the motivation of using diffusion language models (dLLMs) to capture bidirectional dependencies. Our empirical findings provide strong evidence supporting this intuition.
>
> First, we observe that the diffusion LLM, even without the tree search module (w/o MDTS), consistently outperforms autoregressive LLMs (i.e., OPRO) (see Table 2).
>
> Second, the benefit of bidirectional modeling aligns closely with the nature of many black-box design tasks. In domains such as DNA sequence design, the quality of a candidate depends on global sequence context, where a base at position i can influence and be influenced by both its left and right neighbors. Diffusion LLMs naturally model such dependencies by jointly predicting masked tokens conditioned on both sides of the context, in contrast to left-to-right autoregressive generation that enforces a one-way dependency. This mechanism provides a more holistic view of the design space.
>
> > The originality of the paper is somewhat limited. As mentioned, the work feels like a natural and incremental extension of existing ideas. The field already has "LLMs for BBO" and "diffusion models for BBO". This paper's "in-context denoising" component is a straightforward combination of these two, which has been introduced by Tang et al. 2025 to optimize molecules. Similarly, the use of MCTS to guide diffusion models is also an existing area of research, which the authors acknowledge, e.g., Tang et al. (2025, which proposes an MCTS framework built on top of diffusion language models for molecular optimization, more specifically, SMILES. This work, on the other hand, has studied DNA sequences, which appear to be even more approachable than SMILES as SMILES encodes complicated structural information and many atoms.
>
> We thank the reviewer for the comment on originality. While our work builds upon existing directions of using large models for black-box optimization (BBO), it differs substantially from prior approaches in both scope and formulation.
>
> First, prior LLMs for BBO methods primarily address white-box or online optimization by leveraging autoregressive LLMs to generate or refine designs in a task-specific manner. In contrast, our method targets offline BBO, where only a few labeled examples are available and no online feedback can be used, and we are the first to explore diffusion LLMs for this context to model bidirectional dependency on design.
>
> Second, while Tang et al. (2025, Peptune) indeed employ diffusion models for drug design, their model is a domain-specific discrete diffusion model trained from scratch for peptide optimization. It lacks the in-context reasoning and general-purpose knowledge of large diffusion language models. Our approach is the first to use a pre-trained diffusion LLM as a universal generator, capable of operating across heterogeneous domains (e.g., DNA and continuous morphology) without retraining for each task. This enables a single model to perform adaptation across both discrete and continuous design spaces, which is something prior works can not achieve.

---

> ### Author Response · Authors · 2025-11-22
>
> > The ablation study in Table 2 is a major concern. Removing MDTS (the tree search) leads to a massive performance drop (e.g., 0.876 to 0.798 on TF8; 0.642 to 0.503 on TF10). This strongly suggests that the search framework (MCTS guided by the GP's EI) is doing almost all the work. The dLLM is just acting as a proposal mechanism within this search.
>
> We appreciate the reviewer’s comment on the ablation study. While removing the masked diffusion tree search (MDTS) indeed reduces performance, this does not mean that MDTS alone drives the improvement.
>
> Importantly, as shown in Table 2, our "vanilla diffusion" ablated variant replaces the diffusion LLM with a continuous diffusion model, while still using the same MDTS framework. This variant therefore isolates the contribution of the diffusion LLM itself. The large performance gap between “vanilla diffusion” and our full dLLM demonstrates that the pre-trained diffusion LLM provides strong generalization and in-context reasoning advantages, especially in the few-shot regime where we don't have enough data to train a reliable task-specific diffusion model.
>
> Meanwhile, MDTS is not an external add-on, but a search procedure that is tightly coupled with the diffusion LLM through the UCT term and the candidate discrete denoising process. This ensures that the diffusion LLM actively guides exploration and exploitation during search rather than merely serving as a proposal sampler.
>
> In summary, the two modules play complementary roles: the pre-trained diffusion LLM offers a universal generative prior with strong generalization, while MDTS provides a principled mechanism to efficiently explore the design space under uncertainty.
>
> ## Questions
> > To truly isolate the contribution of the dLLM as a better mechanism then AR, could the authors add an ablation study? Specifically, please compare the full dLLM method against a baseline that uses the exact same MDTS framework (same tree depth, branches, GP-EI reward) but replaces the dLLM with a simpler mechanism (e.g., simpler generative models such as a VAE)
>
> We thank the reviewer for the helpful suggestion. In fact, our vanilla diffusion ablation already serves the purpose of isolating the contribution of the diffusion LLM under the same MDTS framework. Specifically, in this variant we replace the diffusion LLM with a continuous diffusion model trained directly on the small offline dataset, while keeping the same MDTS configuration (identical tree depth, branching factor, and GP-EI reward). This directly evaluates whether the improvement comes from the pre-trained diffusion LLM itself rather than from the search procedure.
>
> As shown in Table 2, this vanilla diffusion baseline performs substantially worse than our full method, demonstrating that the pre-trained diffusion LLM contributes significantly to the overall performance gains.
>
> > As mentioned, dLLMs are claimed to be better about "bidirectional dependencies" than LLMS, but no theoretical explanations are given. If we only examine experimental validations, results on TF8/TF10 (DNA sequences) seem to support this, as the gaps to LLMs are largest there. However, it seems performance gaps on the continuous tasks are tiny. Is the method's advantage primarily for discrete, sequence-like design spaces, and less so for continuous ones? Is there a theory to explain this?
>
> We thank the reviewer for the observation. We would like to clarify that the improvements of dLLM are not limited to discrete sequence-like tasks. Our method also achieves clear gains over autoregressive (AR) LLMs on continuous tasks such as Ant Morphology (0.652 versus 0.596) and D’Kitty Morphology (0.942 versus 0.878) (Table 1). While current LLMs are generally more proficient with symbolic alphabets like A/C/G/T than with long floating-point sequences, dLLM still provides consistent and measurable improvements, demonstrating its effectiveness across both discrete and continuous domains.

---

### Meta-Review · Area_Chair_ExCm · 2026-01-07

**Summary:**

This paper introduces dLLM, a framework designed for offline Black-Box Optimization (BBO). The authors argue that traditional autoregressive Large Language Models (LLMs) used for BBO generate designs from left-to-right. This prevents them from capturing bidirectional dependencies—where a middle part of a sequence (like a DNA base) depends on both its left and right neighbors.

My primary concern is that the authors attribute their success to the bidirectional nature of Diffusion LLMs, yet the experimental design does not decouple the generative model from the powerful search framework (MCTS + GP). For example, to prove that diffusion is superior to autoregressive (AR) models for BBO, the authors must provide a comparison against an AR-LLM + MCTS + GP baseline.

**Reviewer Concerns:**

Reviewers (9Unr) noted that "LLMs for BBO," "Diffusion for BBO," and "MCTS for Diffusion" are all existing concepts. They felt the paper is a straightforward combination of these. Pointed out that removing the tree search (MDTS) causes a massive performance drop. This suggests the GP-guided search is doing the "heavy lifting," while the Diffusion LLM is just a "proposal mechanism."

Reviewers (9Unr, 78bn) noted there is no formal theory or concrete example proving that "bidirectional modeling" is actually what makes the model better. Reviewer 78bn highlighted the "curse of dimensionality," questioning how a GP trained on only a few examples can provide reliable guidance in high-dimensional design spaces.

**Reviewer Scores:**

The authors have addressed some of the concerns the reviewers raised, but I don't think the reviewers would be satisfied enough for them to raise they score above the acceptance threshold.

---

### Decision · Program_Chairs · 2026-01-26

Reject